

# Long-term groundwater recharge rates across India by in situ measurements

Soumendra N. Bhanja[1,2], Abhijit Mukherjee[1,3,4], Rangarajan Ramaswamy[5], Bridget R. Scanlon[6], Pragnaditya Malakar[1], Shubha Verma[7]

5  [1]Department of Geology and Geophysics, Indian Institute of Technology Kharagpur, West Bengal 721302, India
[2]Presently at Faculty of Science and Technology, Athabasca University, Alberta T9S3A3, Canada
[3]School of Environmental Science and Engineering, Indian Institute of Technology Kharagpur, West Bengal 721302, India
[4]Hydroscience and Policy Advisory Group, Indian Institute of Technology Kharagpur, West Bengal 721302, India
[5]National Geophysical Research Institute, Uppal Road, Hyderabad 500 007, India
10  [6]Bureau of Economic Geology, Jackson School of Geosciences, The University of Texas at Austin, Texas 78713, USA
[7]Department of Civil Engineering, Indian Institute of Technology Kharagpur, West Bengal 721302, India
[*]Corresponding authors: (soumendrabhanja@gmail.com) and Abhijit Mukherjee

*Correspondence to*: Soumendra N. Bhanja (soumendrabhanja@gmail.com) and Abhijit Mukherjee (amukh2@gmail.com, abhijit@gg.iitkgp.ernet.in)

15  **Abstract.** Groundwater recharge sustains groundwater discharge, including natural discharge through springs and base flow to surface water as well as anthropogenic discharge through pumping wells. Here, for the first time, we compute long-term (1996-2015) groundwater recharge rates using data retrieved from several groundwater level monitoring locations across India (3.3 million km[2] area), the most groundwater-stressed region globally. Spatial variations in groundwater recharge rates (basin-wide mean: 17 to 960 mm/yr) were estimated in the 22 major river basins across India. The extensive plains of the 20  Indus-Ganges-Brahmaputra (IGB) river basins are subjected to prevalence of comparatively higher recharge. This is mainly attributed to occurrence of coarse sediments, higher rainfall, and intensive irrigation-linked groundwater abstraction inducing recharge by increasing available groundwater storage and return flows. Lower recharge rates (<200 mm/yr) in most of the central and southern study areas occur in cratonic, crystalline fractured aquifers. Estimated recharge rates have been compared favorably with field-scale recharge estimates (n= 52) based on tracer (tritium) injection tests. Results show 25  precipitation rates are not significantly influencing groundwater recharge in most of the river basins across India, indicating human influence in prevailing recharge rates. The spatial variability in recharge rates could provide critical input to policy makers to develop more sustainable groundwater management in India.

## 1 Introduction

India represents ~18% of the global population but occupies <3% of the global land area (FAO, 2013). Agricultural activity 30  is intensive and parts of India supports the highest rate of irrigated arable land globally (Siebert et al., 2013). The Ministry of Water Resources, Government of India and the World Bank (1998), estimate that groundwater contributes ~9% to India's GDP. However, few studies have reported groundwater recharge at a limited number of locations (Goel et al., 1975;





Bhandari et al., 1982; Athavale et al., 1992; Rangarajan et al., 1995; Rangarajan et al., 1997; Athavale et al., 1998; Rangarajan et al., 1998; Rangarajan and Athavale, 2000; Scanlon et al., 2010) (Figure 1). Out of the total irrigation-linked freshwater withdrawal, more than 50% is attributed to groundwater in India (CGWB, 2009).

Groundwater recharge can be defined as "the downward movement of water" that reaches the water table (Healy, 2010).

Recharge can occur directly from the infiltration of rainfall, as well as indirectly via irrigation return flows, leakage from surface water bodies, or a combination of the two (Scanlon et al., 2006). In India, large-scale, anthropogenic redistribution of water impacting recharge have occurred through both canal-driven and groundwater-fed irrigation (Mukherjee et al., 2007; MacDonald et al., 2016). In addition, dry-season groundwater-fed irrigation can increase available groundwater storage, enabling increased recharge during the subsequent monsoons (Revelle and Lakshminarayana, 1975; Shamsudduha et al.,

10   2011).

In India, groundwater recharge (Figure 1) is believed to be derived primarily from a continuous spell of rainfall during the monsoon season (extending from June to September); majority of the total annual precipitation (>74%) across this region has been occurring during monsoon (Guhathakurta and Rajeevan, 2008). Precipitation varies substantially across India; southern India also receives a moderate amount of precipitation during the pre-winter months (October-November). Figure 2 showing

annual precipitation patterns between 1996 and 2015. Droughts occurred in 2002, 2004, 2009 and 2014 in India (Figure 2). Precipitation data also reveal distinct spatial variations that range from humid to arid climates. 22 major river basins have been characterized in India (Figure 1; Bhanja et al., 2017a). The largest river basin in the region is the Ganges Basin in terms of area, followed by Indus Basin and Godavari Basin (Figure 1; Bhanja et al., 2017a). Brahmaputra (Basin 2a) and Barak (Basin 2c) basins have the highest annual rainfall occurrence (> 2000 mm/yr; Table 1) while the Indus basin (Basin 1)

receives the minimum amount of rainfall (Table 1). India includes a range of hydrogeological settings (Figure 3a) that vary between highly fertile alluvial formations within Indus-Ganges-Brahmaputra (IGB) basin aquifers and less permeable, fractured rock aquifers in parts of central and southern India. Thus, IGB basin is intensively cultivated leading to comparatively higher rates of groundwater withdrawal.

Recharge data are critical for developing sustainable groundwater management policies in India. Understanding the controls

on groundwater recharge is also valuable for managing recharge. Currently there are almost no restrictions on groundwater development in India. Electricity policies in India actually promote over development of groundwater. Groundwater has played a large role in agricultural production, supported in large part by intensive irrigation and resulting in poverty reduction (Zaveri et al., 2016). However, low recharge rates relative to rates of pumping has resulted in large groundwater level declines, particularly in northwest India (Bhanja et al., 2017b). The region would face groundwater drought if not

managed properly. Bhanja et al. (2017b) showed replenishment of groundwater storage as a function of implementation of sustainable management policies in parts of the study area.

The objective of this study was to estimate spatial variability in groundwater recharge across India and compare with hydrogeologic environments, precipitation, and irrigation intensities to better understand controls on recharge variability. Unique aspects of this study includes the availability of a network of ~19,000 groundwater level monitoring locations with



~2 decades of data (1996 to 2015), the range of hydroclimatic (arid to humid) and hydrogeologic settings (sedimentary to cratonic) sampled, and the varying intensity of irrigation. Recharge estimates were compared with independent field-scale recharge estimates from 52 locations distributed across India to assess the reliability of the regional-scle estimates. Controls on spatial variability of recharge were also assessed, including precipitation and irrigation intensity across the 22 major river

basins (Figure 1). The river basins provide natural subdivisions of the country that reflect varying hydroclimatic, hydrogeoloic, and anthropogenic activity (Figure 1, Table 1).

## 2 Data and Methods

### 2.1 Groundwater level monitoring

Seasonal (i.e. quarterly, four times a year) ground-based monitoring data (n >15000) for 20 years (from 1996 to 2015), were

a part of the Central Ground Water Board's (CGWB, Government of India) groundwater monitoring mission. Most (~85%) of the studied wells represent shallow, unconfined aquifers (CGWB, 2014); however, these aquifers have not been characterized in CGWB reports/products. Initially, groundwater-level data were filtered for attaining a temporally continuous dataset in the study area for each year. The data were collected once during the months of January, May, August and November in individual years (GREM, 1997). Outliers were removed following Tukey's fence approach (Tukey, 1977)

reducing the usable number of monitoring data points to 3468. The annual change in groundwater levels has determined using the highest and lowest groundwater head elevation (highest and lowest groundwater level generally occur in post-monsoon and pre-monsoon times, respectively) for each year. Finally, gridded ($0.1^0 \times 0.1^0$) Δh data were obtained following the ordinary kriging technique.

### 2.2 Groundwater recharge estimation using the Water Table Fluctuation (WTF) method

In the absence of high-resolution observed data of climatic parameters and other controlling factors that could be used in Several challenges associated with groundwater recharge measurement and high spatiotemporal variability, are the two main concerns for the regional-scale recharge estimates (Healy, 2010). In absence of field-scale data to estimate recharge, simpler methods, such as the water table fluctuation (WTF) technique, are more widely used to estimate recharge ($R_g$) (Healy and Cook, 2002). $R_g$ may be defined as,

$$R_g = \Delta h \times S_y \qquad\qquad\qquad\qquad (1)$$

where, Δh represents seasonal changes in groundwater levels.

Specific yield ($S_y$) of the aquifer material, are one of the important component for the recharge estimation. $S_y$ information were retrieved following CGWB (2012) and Bhanja et al. (2016). Mean $S_y$ values are varying between 0.02 and 0.13, varying with identified aquifers. The IGB basins are heavily irrigated with intensive groundwater withdrawals (Figure 3b)

(Supplementary Figure 1) relative to other parts of the country.





### 2.3 Field-scale recharge estimation using tritium injection approach

Tracer in the form of ions, isotope or gases move with water, can be tracked to estimate water infiltration through unsaturated zone (Healy, 2010). Here, tritium injection method have been used to estimate recharge rates in 52 locations across India (Figure 1). The equivalent depth of water of the peak movement can be represented as the total infiltrated water

flux (Healy, 2010). The piston-flow method, which has been a major assumption in tracer movement technique, is linked with the vertically downward movement of water and dissolved solutes without mixing and change in velocity (Healy, 2010). Tritium injection is performed during the pre-monsoon season and the soil core is collected after the monsoon in order to estimate the monsoon recharge rate (Rangarajan et al., 2000). More details on the tritium injection method can be found in Supplementary Information and in Rangarajan et al. (2000).

### 2.4 Non-linear trend analysis

Non-linearity in the data can be represented through Hodrick-Prescott (HP) trend analysis (Hodrick and Prescott, 1997), which separates cyclical components present ($c_t$) in the data ($y_t$) from the trend ($T_t$) after solving the following equations:

$$y_t = T_t + c_t \tag{2}$$

$$\text{Min (T)} \sum_{t=1}^{T} ((y_t - T_t)^2 + \lambda((T_{t+1} - T_t) - (T_t - T_{t-1}))^2 \tag{3}$$

where, $\lambda$ is a constant (Hodrick and Prescott, 1997; Ravn and Uhlig, 2002).

## 3 Results and Discussions

### 3.1 Recharge estimates from the Water Table Fluctuation method

Calculation of $R_g$ based on water level data from 1996 to 2015 exhibits both spatial and temporal variability over the years (Figure 4). In general, the years of lowest groundwater recharge, i.e. 2002, 2004, 2009 and 2014 correspond to years of

lowest precipitation (Figure 2, 4). $R_g$ exceeding 300 mm/yr is found over most regions of the IGB basin (Figure 4, 5). The central and southern India have been subjected to comparatively lower recharge rates (<200 mm/yr). Basin-wide data show highest recharge rates in basins 1, 2a, 2b, 2c, 10, 11, 14 and 20 (Figure 5, Table 1); parts of the basins are composed of alluvial sediments that are also intensively irrigated linked to groundwater withdrawal. The highest annual recharge rate was calculated for 2010 in most basins. High recharge rates in 2010 are attributed to increased space for recharge related to lower

precipitation in 2009 (Figure 2) with 22 out of 26 meteorological subdivisions declared as rainfall deficit in India (NCC, 2013). Linear regression analysis of recharge rates between 1996 and 2015 show simultaneous occurrence of increased and decreased trend in recharge rates in different parts of India (Figure 6). Parts of Ganges basin (basin 2a) have been experiencing rapid declines in recharge rates while parts of western and north-western India are experiencing increase in recharge in the study period (Figure 6). Non-linear trend analysis shows high temporal variation in recharge in all of the



studied basins (Figure 7); maximum recharge amplitudes are found in basins, 2b, 2c, 11 and 14 (Figure 7). The basins are located in comparatively higher precipitation zones of north-east and west-coastal India, respectively.

**3.2 Groundwater recharge estimates as a function of climate, hydrogeology and irrigation**

Comparatively higher rates of precipitation partly explain the high recharge rates in IGB basin. Precipitation data show high annual variability in all of the basins (Figure 8b). Highly fertile sedimentary formations of IGB basin facilitate both direct and indirect recharge. Higher agricultural groundwater withdrawal in the IGB basin (Figure 3b) leads to decreases in water storage, which can result in increased recharge by generating more recharge space. Subsequently, recharge rates are not homogenous through the IGB basins (Figure 4, 5). Basin-wide mean recharge rates are found to be variable over the years (Figure 8b). Highest inter-annual variability has been obtained in basins 2b, 2c and 14 (Figure 8b); the basins are also experiencing highest precipitation rates (Table 1, Figure 8a). On the other hand, basins located in Indian craton, i.e. basins 3, 4, 5 and 19, exhibit lowest inter-annual variability (Figure 8b). Lower recharge rates are found in central and southern parts of India (Figure 4, 5). The crystalline aquifers in these regions (Mukherjee *et al.,* 2015) are not conducive enough for precipitation-based infiltration through subsurface. The observation is consistent with Sukhija *et al.* (1996), who also found lower recharge rate in fractured regions. Deeply weathered, lateritic soils that have developed on cratons often have low matrix permeabilities due to concentration of kaolinite and development of ferricretes/alucretes (laterites) (Taylor and Howard*,* 1999). These low matrix permeabilities promote infiltration via discontinuities. Soil permeabilities are substantially less than those of the sorted alluvial soils in the IGB basin.

**3.3 Comparison with field-scale recharge estimates**

We present field-scale recharge rates ($R_n$, Figure 1) derived from tritium injection approach (Rangarajan *et al.*, 2000) in Table 2. The recharge rates were estimated in four different land use types, i.e., granite, basalt, sediment and alluvium. $R_n$ varies within a range of 22-179 mm in locations within granite setting and the values reach 46-171 mm in basalt regions (Table 2 and Figure 9). Recharge rates are found to be comparatively higher in sediment and alluvial region with rates of 29-181 and 20-198 mm, respectively (Table 2 and Figure 9). Similarly, $R_g$ also exhibit higher recharge in sediment and alluvium regions (Figure 9). Precipitation rates provide mixed response i.e. alternate high and low values in all of the land-use types.

$R_g$ from this study compare favorably with site-scale determinations in 42 out of the total 52 locations (Figure 9). However, $R_g$ was found to be larger than the recharge rates in location IDs 44, 45, 49, and 50 in Figure 1; Figure 9). Three locations out of four are located in IGB basin, which is experiencing intensive groundwater abstraction for irrigation (Figure 3b). This discrepancy may be coming from the assumption of direct recharge only through the unsaturated zone traced by radioisotopes (Healy, 2010). Also, $R_n$ includes monsoon recharge only, while $R_g$ includes both monsoon and pre-monsoon recharge that is dominated by irrigation return flow. Hence, contributions from indirect recharge via irrigation return flows and recharge amplified by dry-season abstraction for irrigation are ignored in $R_n$ estimates. $R_g$ was found to be lower than $R_n$ in some of the locations (location numbers 10, 11, 25, 28, 40, 46 in Figure 1; Figure 9). All of the six locations are subjected




to comparatively lower rates of irrigation-linked groundwater withdrawal (Figure 3b). Therefore, irrigation return flow and creation of additional recharge space are almost negligible in these locations. Further, the locations are also experiencing comparatively higher rates of precipitation (Table 2). As $R_g$ will not consider a fraction of recharge particularly during aquifer full condition (Healy and Cook, 2002), it is the major reason for the observation of lower $R_g$ values than $R_n$ values in

these six locations.

### 3.4 Groundwater recharge as a function of precipitation

Recharge rates are significantly (*p value* <0.01) correlated with precipitation in 10 out of the 22 basins. Non-linear trend analysis between basin-wide recharge rates and precipitation show good match in basins 2a, 3, 4, 12 and 20 (Figure 7). In contrast, recharge and precipitation trends are negatively correlated in basins 2b, 2c, 7, 16 and 18 (Figure 7). In order to

study the relationship in more detail, we used the Granger causality analysis (Granger, 1988). Results show precipitation significantly (*p value* <0.01) causes $R_g$ in 6 basins, i.e. 8, 11, 12, 13, 15 and 16. Most of these basins are not intensively irrigated; groundwater withdrawal rates are found to be lower in these regions, irrigation groundwater withdrawal found to be less than 75 mm in all of the basins (Figure 3b). Therefore, natural processes i.e. precipitation still influence recharge rates in those basins. Alternatively, the relationship between recharge and precipitation is statistically insignificant in other

basins, which are more intensively irrigated (50 mm to more than 300 mm; Figure 3b). As a result, precipitation influence is found to be less dominant on recharge in these regions.

### 3.5 Assumptions and limitations

Minimal reliance on assumptions in measurement is an advantages of using the WTF method (Healy and Cook, 2002). Furthermore, the method can be followed to estimate recharge for a large region, simultaneously (Healy and Cook, 2002).

Uncertainty in groundwater storage co-efficients (*e.g.* $S_y$) influences the magnitude of computed recharge. In WTF method, recharge rates exhibits minimum recharge values in groundwater discharge regions. The antecedent recession during peak groundwater level (Healy and Cook, 2002), which is a component of base flow and discharge, is difficult to measure accurately and hence results in underestimation of recharge during that time.

### 4 Conclusions

Groundwater recharge is computed from in situ observations between 1996 to 2015 in 22 major river basins across India. Differential spatial patterns of climate, geology and withdrawal linked to irrigation, are the major reasons for spatial heterogeneity in groundwater recharge in India. Higher values (>300 mm/yr) of groundwater recharge rates are observed in alluvial plains of northern and eastern India. Comparatively higher rates of precipitation, high permeability and intense groundwater abstraction, either alone or in combination, are the major reasons. On the other hand, in central and southern

India, comparatively lower recharge occurrence is attributed to lower permeability fractured-crystalline cratonic rocks.





Comparatively lower precipitation and lower irrigation rates are also influencing recharge rates in those regions. Recharge rates based on WTF compare favorably with independent recharge estimates from tracer data in 42 out of 52 locations. Controls on recharge include precipitation in less intensively irrigated basins. Results from this study should provide valuable input to policy makers developing more sustainable groundwater management plans.

**Acknowledgments and Data**

SNB acknowledges CSIR (Government of India) for their support for providing the SPM fellowship. SNB also acknowledges U.S. Department of State for the Fulbright fellowship. We acknowledge CGWB, India and IMD, India, for the water level data and precipitation data, respectively. We thank Robert C. Reedy, UT Austin, for his help in improvement of the figures. We thank Charudutta M. Nirmale for his help in data retrieval.

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





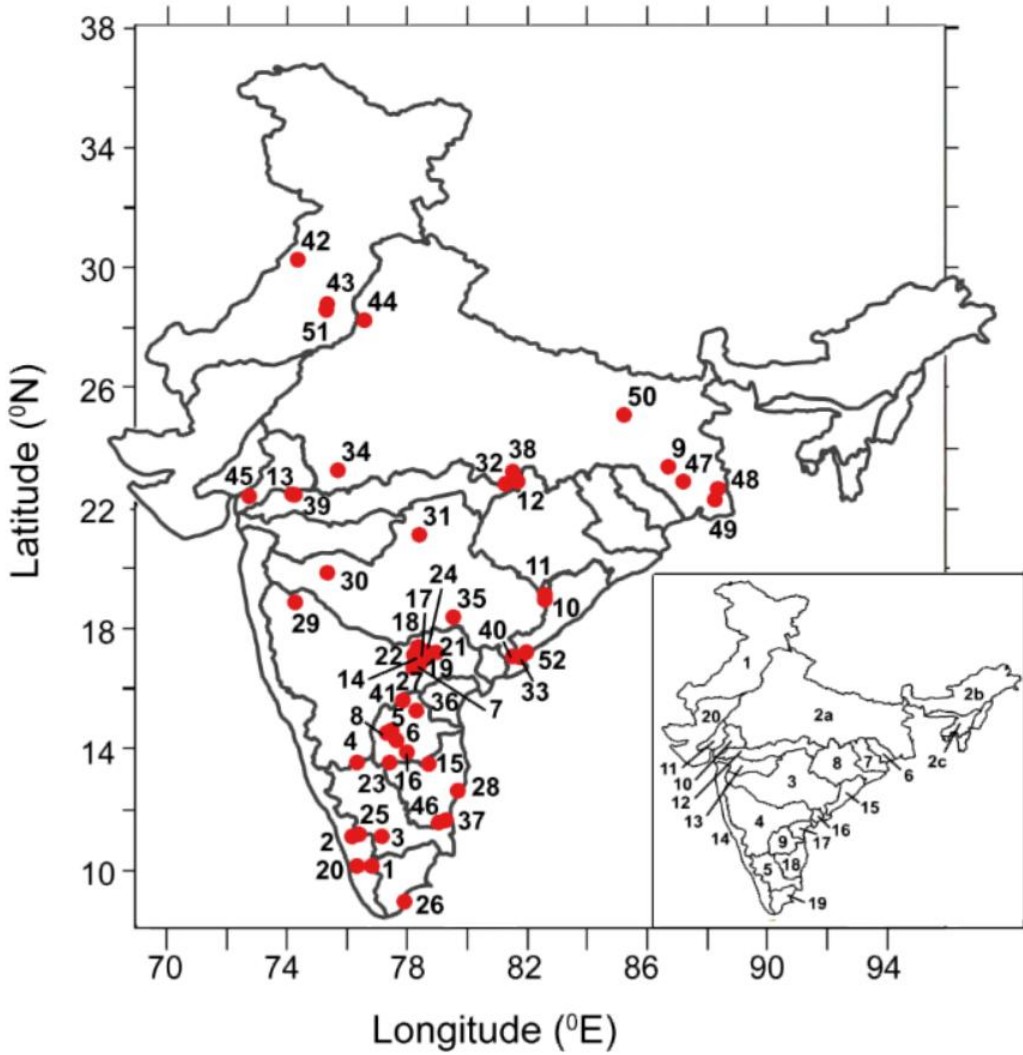

Figure 1: Map of India showing the 22 major river basins and locations (circles) of 52 field-scale recharge measurements used in this study derived from the tritium injection; numbers beside these locations correspond to sites reported in Table 1. River basin numbers are shown in inset. Field-scale recharge data are taken from the following studies and unpublished estimates: Goel *et al.*, 1975; Athavale *et al.*, 1998; Rangarajan and Athavale, 2000; Bhandari *et al.*, 1982; Rangarajan *et al.*, 1997; Rangarajan *et al.*, 1995; Athavale *et al.*, 1992; Rangarajan *et al.*, 1998; Rangarajan *et al.*, 2000.





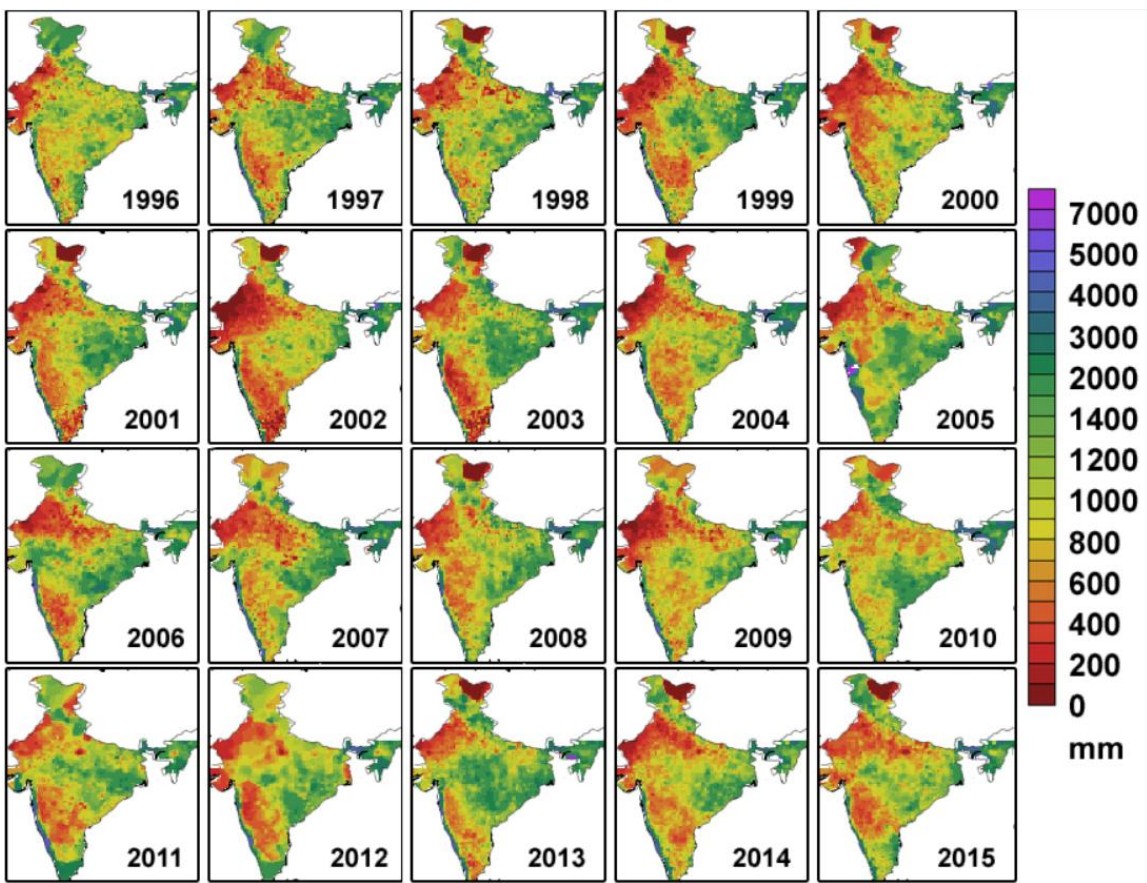

**Figure 2: Maps showing total annual precipitation (mm) over the study area in 1996 to 2015.**




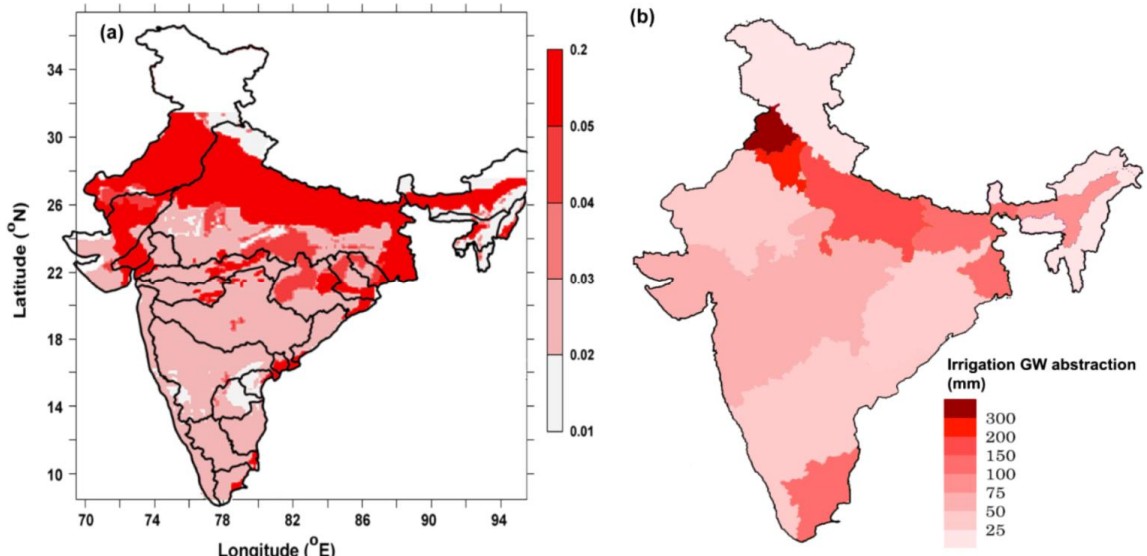

**Figure 3: Maps of (a) specific yield data (modified from Bhanja *et al.*, 2016). Blank area represent areas of no data availability; (b) state-wide groundwater abstraction for irrigation (mm, estimates are for the year 2009) (CGWB, 2012a).**

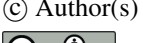



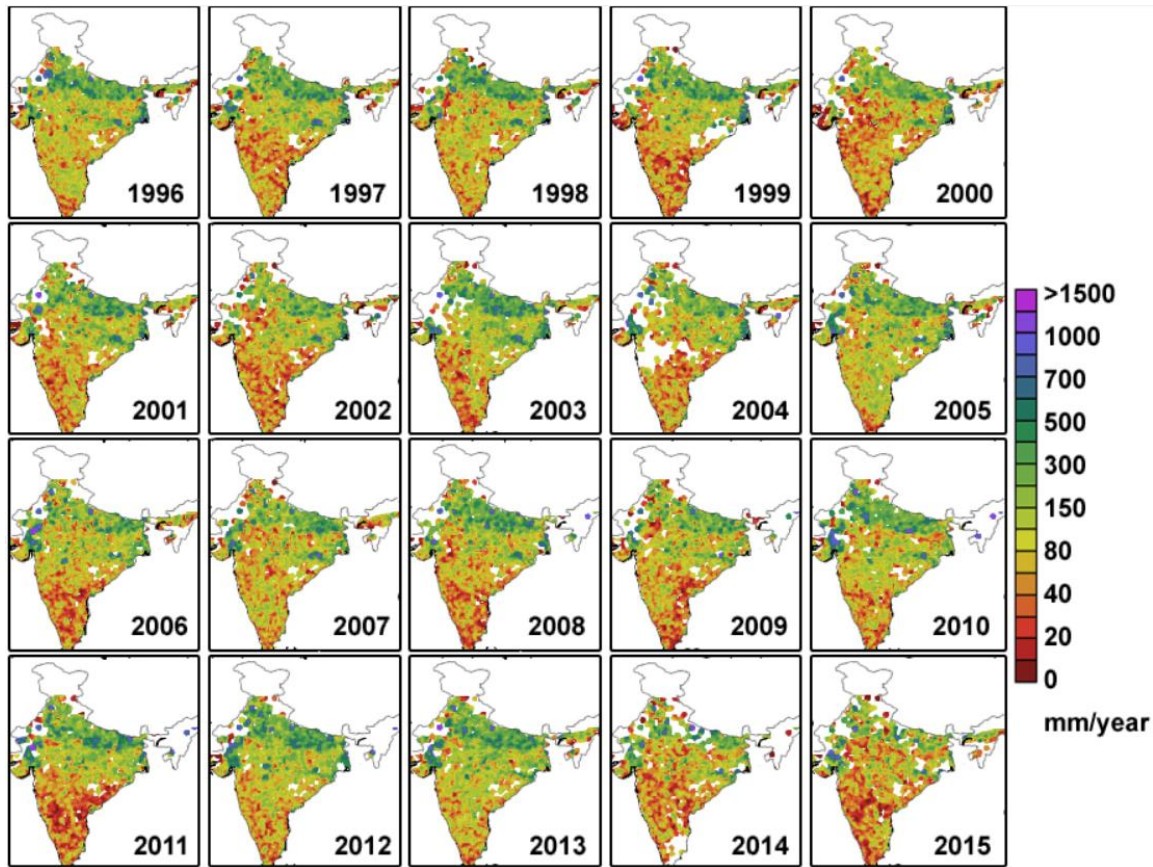

**Figure 4: Maps showing gridded ($0.1^0 \times 0.1^0$) groundwater recharge (mm/yr) calculated using WTF method. Blank area represent areas of no data availability.**





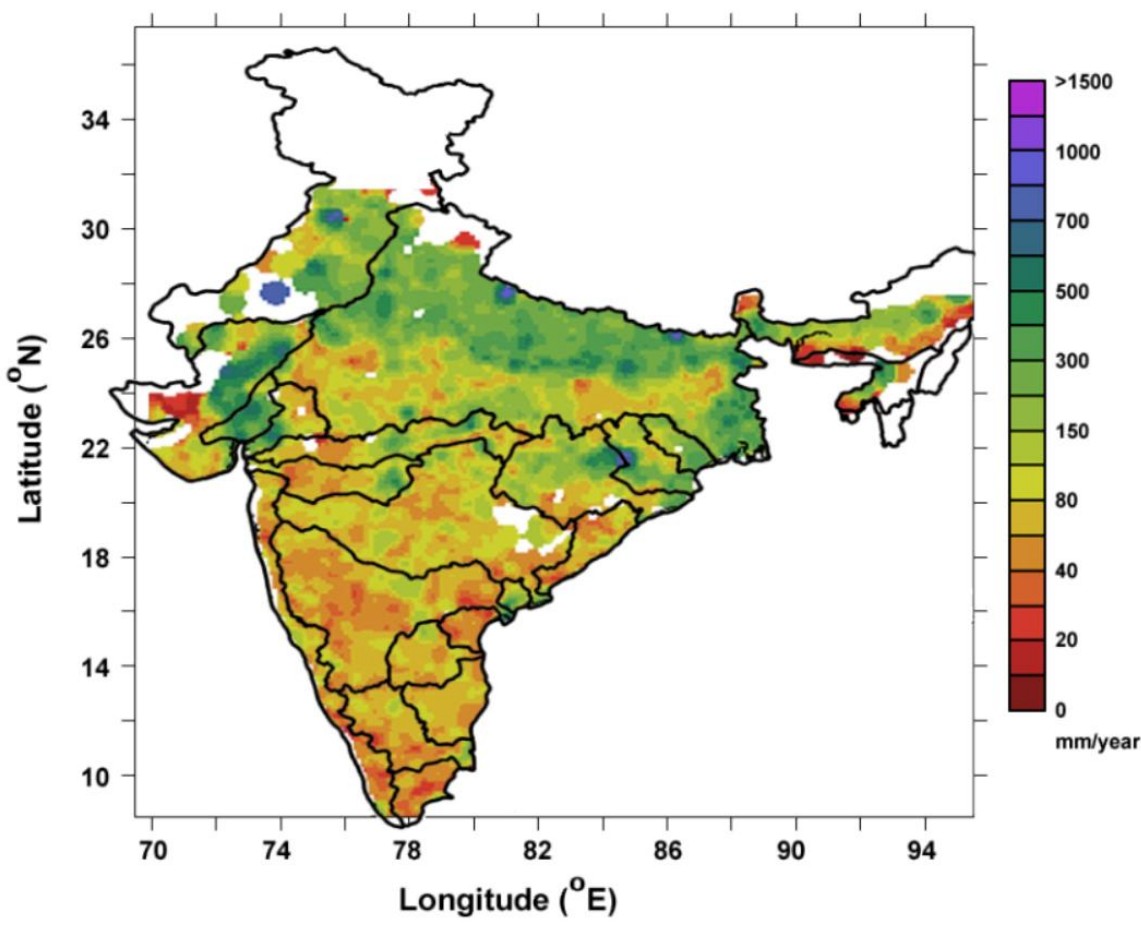

Figure 5: Map of mean recharge in the study period (1996-2015). Blank area represent areas of no data availability.





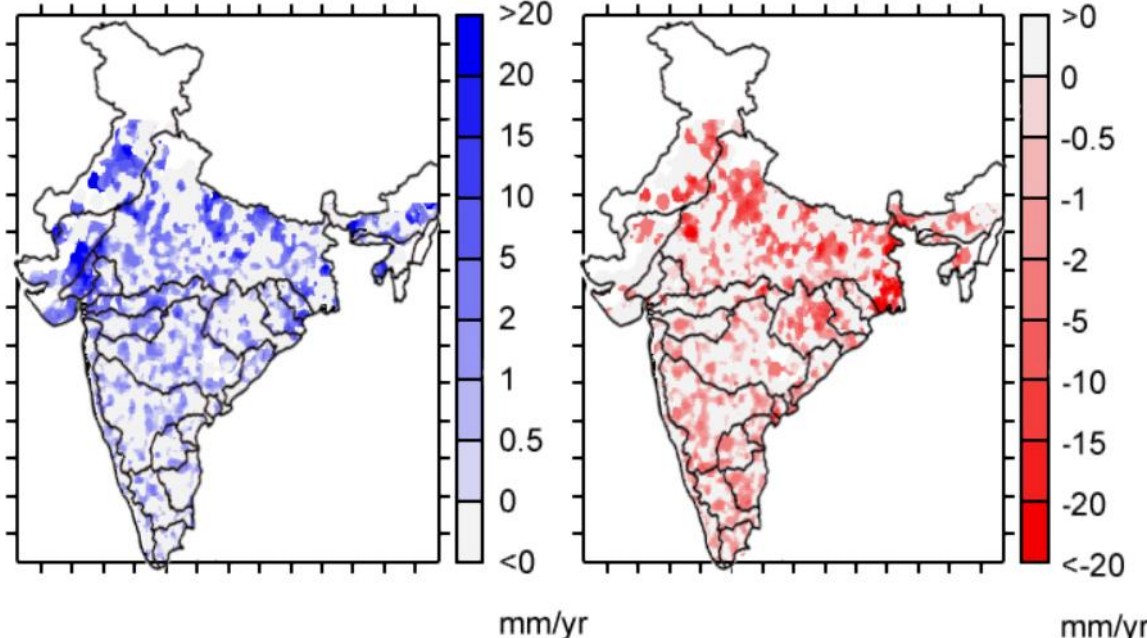

**Figure 6: Maps of positive and negative trends in groundwater recharge in 1996-2015. Basin boundaries are overlaid. Blank area represent areas of no data availability.**





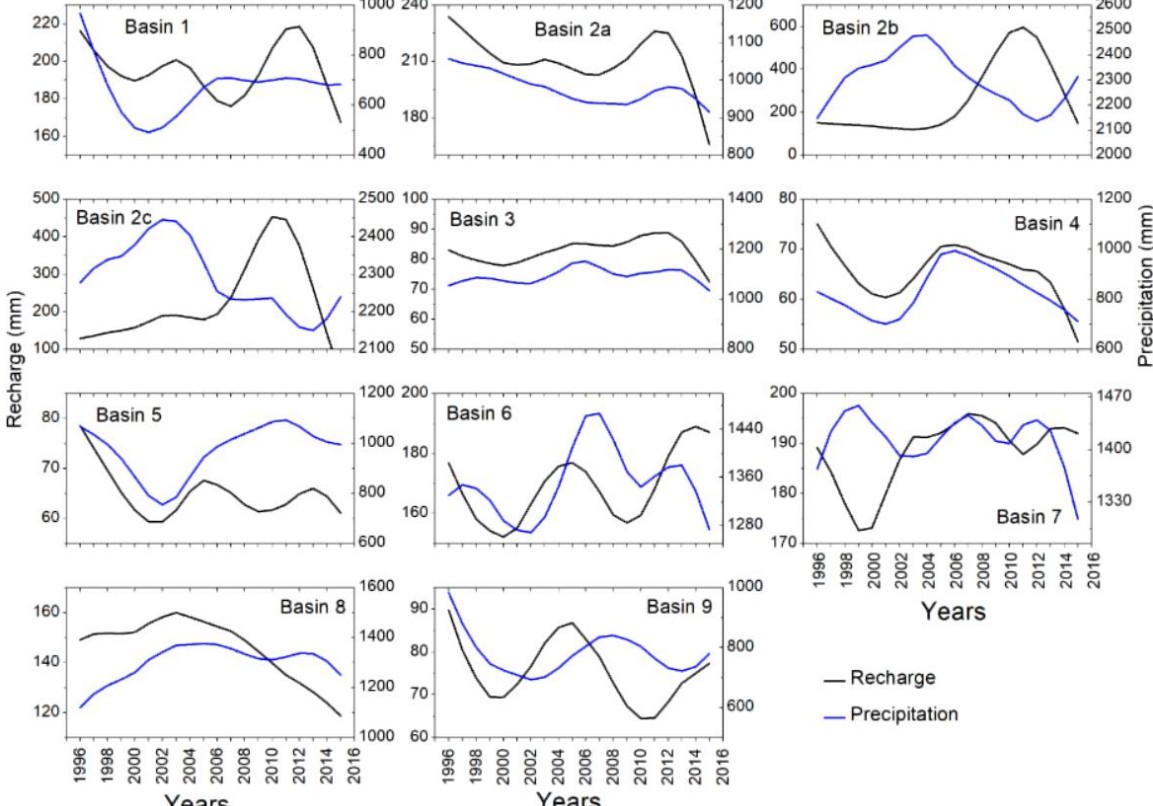



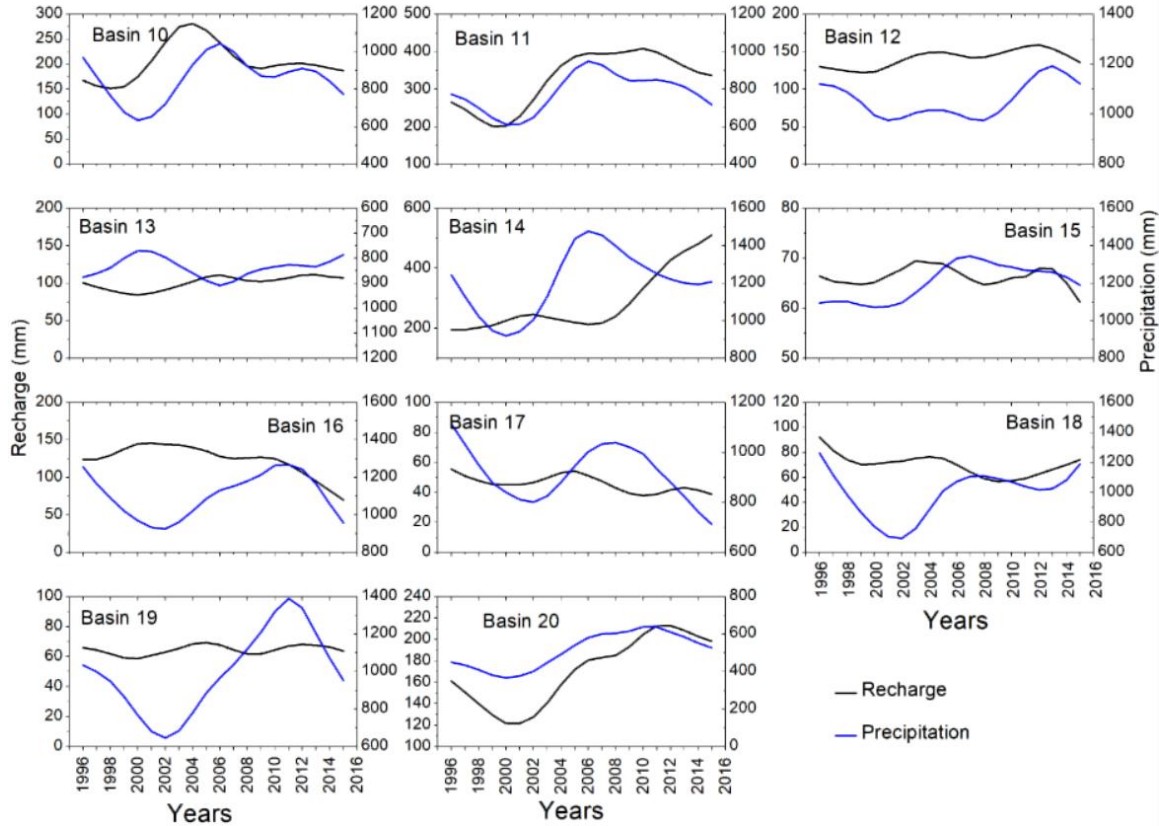

**Figure 7: Basin-wide estimates of Hodrick-Presscott trend analysis of R$_g$ (mm/yr) and precipitation (mm/yr). For basin locations, see Figure 1.**





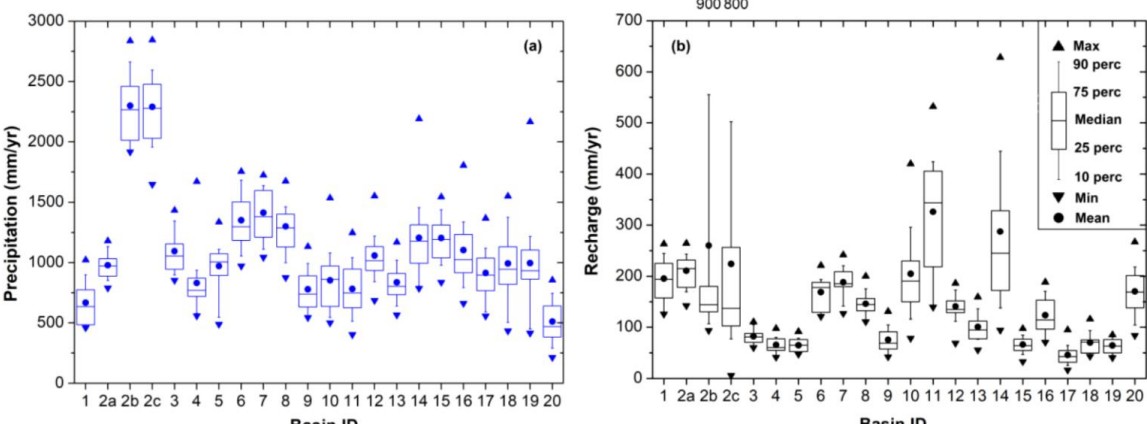

**Figure 8: Basin-wide box-whisker plot of (a) precipitation and (b) recharge rates. Symbols representing mean, median, maximum value (Max), minimum value (Min), 10th percentile, 25th percentile, 75th percentile and 90th percentile are shown in inset of the bottom figure. Recharge estimates exceeded 700 mm/yr data were indicated on the top of the column.**




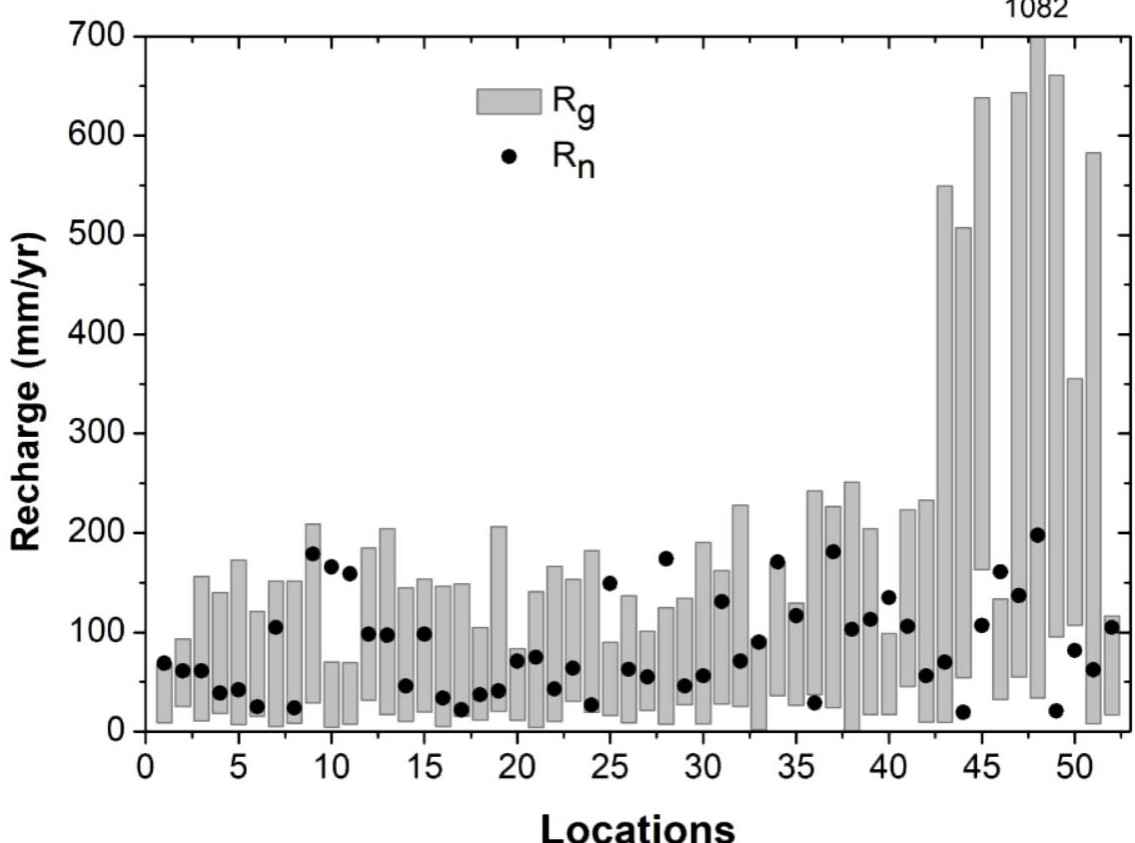

**Figure 9: Comparison of groundwater recharge rates obtained from the present study (annual range) and some of the earlier studies conducted over the point locations shown in Figure 1. Recharge estimates exceeded 700 mm/yr data were indicated on the top of the column.**

