# Peer review of "Long-term groundwater recharge rates across India by in situ measurements"

_Hydrology and Earth System Sciences, 2018_

## Referee Comment (RC1) · Anonymous Referee #1 · 17 Jul 2018

Very interesting paper and should be published but I have some minor comments that should be addressed

(1) Figure 6 – why not show recharge gain and loss on the same figure? (2) Figure 9 – The Rn corresponds to some limited time period...can you compare the same time period for your estimation rather than provide the ranges? (3) How does this relate to the GRACE estimates in Rodell paper? I recall seeing huge losses of water in Northeast India (Punjab) – are there similarities or differences in the loss/gain of groundwater? (4) Figure 8 – can you make it bigger and put recharge and precipitation on the same plot?

[Figure]

313, 2018.

---

## Referee Comment (RC2) · Anonymous Referee #2 · 7 Aug 2018

1.19% Plagiarised document was found it can be reduced. 2. There are various controlling factors in Indian context which are required to be considered for the recharge potential calculation. 3.Hydraulic conductivity, Transmissivity, Rainfall and Recharge variability needed to be calculated. 4.The paper can be resubmitted

---

## Short Comment (SC1) · 7 Aug 2018

Very interesting paper about the recharge rates in India, but I have some suggestions:

1. The manuscript gives comprehensive recharge estimates as a function of climate, hydrogeology, irrigation, and precipitation, but the recharge rates as a function of lithology (particularly carbonates) and slope/altitude is lacking. We know that snow and glacier melt in Himalayan river system provide consistent recharge in higher altitudes.

2) Are these recharge rates applicable to shallow unconfined aquifers only?

---

## Author Comment (AC1) · 9 Sep 2018

*"Long-term groundwater recharge rates across India by in situ measurements"*, by Soumendra N. Bhanja, Abhijit Mukherjee, Rangarajan Ramaswamy, Bridget R. Scanlon, Pragnaditya Malakar, Shubha Verma

**General Comment from Authors and highlights of revision:**

Following the suggestion of the Reviewers, we have done a complete revision of the manuscript. We have diligently tried to respond to all of the reviewer concerns in the previous version of the manuscript, the responses are stated below. We believe, the manuscript have improved to a great respect.

In summary, we have:

1. Inserted a new hydrogeology map, Figure 3

2. Included more discussions based on the reviewers' suggestions

3. Modified several figures based on the reviewers' concern

**Reviewer #1**:

Very interesting paper and should be published but I have some minor comments that should be addressed.

Reply: We would like to thank the reviewer for his/her appreciation.

**Rev 1. Comment 1:** Figure 6 – why not show recharge gain and loss on the same figure?

Reply: Following the reviewer's concern, we have modified the Figure 6.

[Figure]

mm/yr

Figure 6: Maps of positive and negative trends of groundwater recharge in 1996-2015. Basin boundaries are overlaid. Blank area represent areas of no data availability.

**Rev 1. Comment 2:** Figure 9 – The Rn corresponds to some limited time period: : :can you compare the same time period for your estimation rather than provide the ranges?

Reply: Some of the Rn values are measured before the study period (Rangarajan et al., 2000). Due to scarcity of Rn data availability, we have provided the range of the recharge rates obtained through the WTF approach in order to show the magnitudes of the two different recharge estimates.

**Rev 1. Comment 3:** How does this relate to the GRACE estimates in Rodell paper? I recall seeing huge losses of water in Northeast India (Punjab) – are there similarities or differences in the loss/gain of groundwater?

Reply: Comparing groundwater recharge rates with GRACE-based storage is out of the scope of this study. Details regarding in situ and satellite-based groundwater storage could be found in Bhanja et al. (2016, 2017, 2018), in two of them Dr. Rodell is also a co-author.

**Rev 1. Comment 4:** Figure 8 – can you make it bigger and put recharge and precipitation on the same plot?

Reply: We would like to thank the reviewer for his/her comment. Following the reviewer's suggestion, we have modified the Figure 8 to include precipitation and recharge both.

[Figure]

Figure 8: Basin-wide box-whisker plot of precipitation (blue) and recharge rates (black). Symbols representing mean, median, maximum value (Max), minimum value (Min), 10th percentile, 25th percentile, 75th percentile and 90th percentile are shown in inset of the bottom figure. Recharge estimates exceeded 700 mm/yr data were indicated on the top of the column.

References

Bhanja, S. N., A. Mukherjee, M. Rodell (2018). Groundwater storage anomaly in India. In: A. Mukherjee Ed. Groundwater of South Asia. Springer Verlag publishing house.

Bhanja, S. N., A. Mukherjee, M. Rodell, Y. Wada, S. Chattopadhyay, I. Velicogna, K. Pangaluru, and J. S. Famiglietti (2017). Groundwater rejuvenation in parts of India influenced by water-policy change implementation. Scientific Reports, 7, 7453.

Bhanja, S. N., A. Mukherjee, D. Saha, I. Velicogna, and J. Famiglietti (2016). Validation of GRACE based groundwater storage anomaly using in-situ groundwater level measurements in India. Journal of Hydrology, 543(B), 729–738.

---

## Author Comment (AC2) · 9 Sep 2018

*"Long-term groundwater recharge rates across India by in situ measurements"*, by Soumendra N. Bhanja, Abhijit Mukherjee, Rangarajan Ramaswamy, Bridget R. Scanlon, Pragnaditya Malakar, Shubha Verma

**General Comment from Authors and highlights of revision:**

Following the suggestion of the Reviewers, we have done a complete revision of the manuscript. We have diligently tried to respond to all of the reviewer concerns in the previous version of the manuscript, the responses are stated below. We believe, the manuscript have improved to a great respect.

In summary, we have:

1. Inserted a new hydrogeology map, Figure 3

2. Included more discussions based on the reviewers' suggestions

3. Modified several figures based on the reviewers' concern

**Reviewer #2**:

**Rev 2. Comment 1:** 19% Plagiarised document was found it can be reduced.

Reply: We respectfully disagree with the reviewer's concern. The present version of the manuscript is checked with state-of art similarity diagnostic software, and only negligible similarity is found. Even the similar texts are technical words/phrases or nouns or our generic writing in previous publications. We believe the HESS Editorial Office has also done a thorough check and are now satisfied with the level of any potential similarity.

**Rev 2. Comment 2:** There are various controlling factors in Indian context which are required to be considered for the recharge potential calculation.

Reply: We have not studied the recharge potential calculation, we have computed recharge estimates from field data. We have also compared the recharge estimates from two different estimates.

**Rev 2. Comment 3:** Hydraulic conductivity, Transmissivity, Rainfall and Recharge variability needed to be calculated.

Reply: We would like to thank the reviewer for his/her concern. We have added a paragraph in Section 3.2.

[revised manuscript text omitted]

**Rev 2. Comment 4:** The paper can be resubmitted

Reply: We are looking forward to resubmit.

---

## Author Comment (AC3) · 9 Sep 2018

*"Long-term groundwater recharge rates across India by in situ measurements"*, by Soumendra N. Bhanja, Abhijit Mukherjee, Rangarajan Ramaswamy, Bridget R. Scanlon, Pragnaditya Malakar, Shubha Verma

**General Comment from Authors and highlights of revision:**

Following the suggestion of the Reviewers, we have done a complete revision of the manuscript. We have diligently tried to respond to all of the reviewer concerns in the previous version of the manuscript, the responses are stated below. We believe, the manuscript have improved to a great respect.

In summary, we have:

1. Inserted a new hydrogeology map, Figure 3

2. Included more discussions based on the reviewers' suggestions

3. Modified several figures based on the reviewers' concern

**Prof. Ghulam Jeelani's comments**:

Very interesting paper about the recharge rates in India, but I have some suggestions:

Reply: We would like to thank Prof. Jeelani for his encouraging comments.

**SC 1. Comment 1:** The manuscript gives comprehensive recharge estimates as a function of climate, hydrogeology, irrigation, and precipitation, but the recharge rates as a function of lithology (particularly carbonates) and slope/altitude is lacking. We know that snow and glacier melt in Himalayan river system provide consistent recharge in higher altitudes.

Reply: Carbonate aquifers are mostly confined within the Indian state Jammu and Kashmir, the region is kept outside the study region because of limited data availability. We would consider these in future studies.

**SC 1. Comment 2:** Are these recharge rates applicable to shallow unconfined aquifers only?

Reply: We agree with Prof. Jeelani, these recharge rates are computed for shallow unconfined aquifers only. We don't have enough data availability for computing recharge rates in different types of aquifers.

---

## Editor Decision (ED1)

**Long-term groundwater recharge rates across India by in situ measurements**

[revised manuscript text omitted]

**3.3 Comparison with field-scale recharge estimates**

We present field-scale recharge rates (R$_n$, Figure 1) derived from tritium injection approach (Rangarajan *et al.*, 2000) in [3] Table 2. The recharge rates were estimated in four different land use types, i.e., granite, basalt, sediment and alluvium. R$_n$ varies within a range of 22 to 179 mm in locations within granite setting and the values reach 46 to 171 mm in basalt regions (Table 2 and Figure 10). Recharge rates are found to be comparatively higher in sediment and alluvial region with rates of 29-181 and 20-198 mm, respectively (Table 2 and Figure 10). Similarly, R$_g$ also exhibit higher recharge in sediment and alluvium regions (Figure 10). [4] Precipitation rates provide mixed response i.e. alternate high and low values in all of the land-use types.

Number: 1 Author:     Subject: Inserted Text          Date: 12/21/18, 1:02:25 PM

s

Number: 2 Author:     Subject: Inserted Text          Date: 12/21/18, 1:02:19 PM

Number: 3 Author:     Subject: Highlight      Date: 12/21/18, 1:04:27 PM

Table missing.

Number: 4 Author:     Subject: Highlight      Date: 12/21/18, 1:11:20 PM

Rewrite sentence to clarify. Do you mean variable ppt produces variable recharge?

[revised manuscript text omitted]

Number: 1 Author: Subject: Highlight Date: 12/21/18, 1:08:41 PM
Sentence does not make sense. Looks like all of the Max symbols exceed 700. "exceeded" should be changed to "exceeding" unless I am misunderstanding meaning of the sentence.

[Figure]

**Figure 10: Comparison of groundwater recharge rates obtained from the present study (annual range) and some of the earlier studies conducted over the point locations shown in Figure 1. Recharge estimates exceeded 700 mm/yr data were indicated on the top of the column.**

Number: 1 Author:    Subject: Inserted Text    Date: 12/21/18, 2:01:06 PM
ing

Number: 2 Author:    Subject: Inserted Text    Date: 12/21/18, 2:01:17 PM
are